# Determination and quantitative description of hollow body in point cloud

**Abstract**

When volume 3d display system deals with point cloud data with hollow bodies, the hollow body area cannot be determined correctly causing the lack of color information in responding area. Existing researches lack a solution to determine hollow bodies. This paper firstly gives a quantitative description of hollow body and defines a set of parameters to describe size,shape,position of hollow bodies. Then this paper proposes a voxel connectivity regional-growth hollow body determination algorithm(VCRHD) to determine the hollow bodies in 3D point cloud. The algorithm has two phases. The first phase is to use a small amount of voxels to realizes the voxelization of the point cloud and calculate the approximate volume ratio of hollow bodies based on the voxel connectivity regional-growth principle. Then this paper uses the approximate volume ratio to determine the optimal number of voxels based on the experimental result. The second phase is to use the optimal number of voxels to determine the hollow bodies and calculate hollow body parameters which is proved to be efficient and accurate. In addition,this paper establishes a data set containing 287 different point cloud files in 7 different categories to test algorithm. The experimental results prove the feasibility of the algorithm. Finally,this paper analyzes the limitations of the algorithm and looks forward to the application prospect in the future.

**Index Terms:** Computing methodologies—Volumetric models

## 1 Introduction

With the development of computer graphics,digital image processing and other technologies,people demand more realistic display effects. In real world, objects have 3D physical dimensions containing length,width and height. However,the majority of display technologies remain 2D display technologies that satisfy some psychological depth cues but lack depth information,which will lead to degeneration of the functionalities of the human eyes' depth,dynamics,and parallax [1]. The human brain need to acquire information completely consistent with the 3D real object which is called 3D information.

Volumetric 3D display enables a range of cues to depth to be inherently associated with image content and permits images to be viewed directly by multiple simultaneous observers in a natural way [2]. So volumetric 3D display is a significant research direction of true 3D display. Recent years there are many researches about volumetric 3D display. In order to solve the problem of large-scale 3D display,Xiaoxing Yan [3] and Yuan Liang [4] propose to construct a 3D entity sandbox display system. They firstly extract DEM data from point cloud and then build the the sandbox model by means of the mechanical structure. Then they install LEDs on the mechanical structure to visually display various 3D terrain,architecture and other scenes. Jin su Lee,Mu young Lee,et al. [5] design a stereoscopic 3D monitor based on a periodic point source (PLS) structure using a multifocal lens array (MFLA). They construct point lights emitted by the multifocal lens array to form voxels of discrete 3D images in the air. Jun Sun [6] studies a research on true three-dimensional dynamic display algorithm of the array LED three-dimensional display system and builds an array LED three-dimensional display system.

However,such 3D display systems don't do well when processing data with hollow bodies such as building gates and bridges. When data is transmitted to the system for display,the voxels used to display hollow bodies won't receive any color information because these area have no points. So it can't reproduce the original 3D scene and will seriously affect the effect of the 3D display system. Therefore,if we can accurately find each hollow body in the point cloud and use the voxels of hollow bodies to simulate the color that the human eye should observe through them, we can achieve a more realistic three-dimensional display. Besides,the determination of hollow bodies can help in CAD and surface construction.

This paper proposes the concept of hollow body and studies how to solve the problem of hollow bodies in point cloud. The contributions of the paper are as follows:

-This paper proposes the concept of hollow body. Then this paper gives a quantitative description of hollow body and defines a set of parameters to describe its characteristics.

-This paper proposes a novelty voxel connectivity regional-growth hollow body determination algorithm to determine the hollow bodies in point cloud. The algorithm has two phases. The first phase is to use a small amount of voxels to calculate approximate values of volume ratio of hollow bodies. Then this paper uses the approximate value to determine the optimal number of voxels based on our experimental result. The second phase is to use the optimal number of voxels that divide the point cloud to determine hollow bodies and hollow body parameters accurately and efficiently.

-Considering the concept of hollow body is proposed for the first time,there aren't related experimental results to make comparison. Therefore,this paper establishes a dataset to verify the effect of the algorithm and provides a reference for subsequent work.

## 2 Related works

There are similar researches to determine the boundary of 3D point cloud and holes. The extraction methods of boundary in 3d point cloud can be divided into two categories:methods based on grids and directly on points. The methods based on the grids can identity the boundary through the topological relationship between the grids [7–12]. Yongtae Jun [8] firstly determine the seed boundary point by the principle that one of the adjacent three triangles of the boundary triangle mesh is empty and then obtain the closed loop boundary edge by tracking. Xianfeng Huang,Xiaoguang Cheng et al. [10] construct TIN grids for point cloud. Then they initialize a maximum boundary and gradually narrow the boundary edge by setting the

threshold of the length of edge. The methods directly based on points firstly find the neighbors of each point and then calculate the geometric properties of points such as normal vector,density to determine the boundary [13–18]. Van Sinh Nguyen,Trong Hai Trinh,Manh Ha Tran [14] project the point cloud into the two-dimensional grid on the xy plane and obtain the point cloud boundary based on the number of points in each grid and their adjacent grids. Fan LU,Song Li et al. [18] project the point cloud on the plane and use the inverse distance weight and the point cloud density to determine boundary feature points. Shaoyan Gai,Feipeng Da,Lulu Zeng,Yuan Huang [17] use a 2d phase map and the adjacent area of each point to determine the boundary. Then they use the row and column coordinates of the boundary point to remove the contour points.

However,existing researches lack a solution to deal with the determination of hollow bodies. They can only determine the boundary of point cloud or the holes. So this paper gives a quantitative description of hollow body and proposes the VCRHD algorithm to determine of hollow bodies.

## 3 Hollow body

The hollow body is an empty area which is surrounded by points and has points above it. Besides it must connect with the outer area of the model. We set y-axis as the height axis,so the hollow body is an independent area that is closed to the upper area in y-axis and connects with the outer area in x-axis or z-axis direction. The regions that have no point or a small number of points may belong to hollow bodies. As shown in figure 1,the areas beyond the green boundaries are six hollow bodies. In this paper,we propose the VCRHD algorithm to find each hollow body in the point cloud and define volume ratio,normal line,depth ration to describe their size,position and shape.

### 3.1 Determination of the hollow body

#### 3.1.1 Overview

The algorithm has two phases. The process of the two phases are similar. The propose of the first phase is to use a small amount of voxels to divide the point cloud and calculate approximate values of volume ratios of hollow bodies. The approximate value of volume ratio is used to calculate the optimal number of voxels dividing the point cloud which is more accurate and efficient based on the experimental result. In the second phase,the algorithm uses the optimal number of voxels to determine the hollow bodies and calculate the hollow body parameters accurately.The process of the algorithm is as figure 2.

#### 3.1.2 Voxelization of point cloud

After obtaining the preprocessed point cloud data,we first rotate it through meshlab to adjust y-axis as its height axis and then implement voxelization of the point cloud according to the following rules.

Step 1:Determine segmentation parameter of voxelization
Calculate the range of point cloud in x,y,z axis as range_X,range_Y,range_Z and define min as the minimum of the three number. Meanwhile,define k as division parameter.$d_t$=min/k. We find that the number of voxels used to achieve the same accuracy is different for point cloud data containing different volume ratios of hollow body. So we summarize the rule in the experimental part. Therefore k in the first phase is to ensure the model area of point cloud is divided into 24000 voxels which is the experimental result. And in the second phase we adjust k

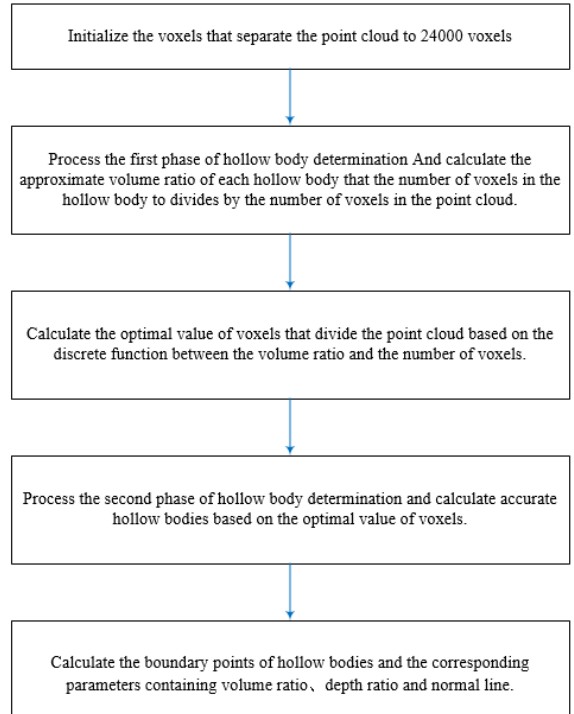

Figure 2: Process of the VCRHD algorithm

to divide point cloud based on the optimal number of voxels.

Step 2: Define plane matrix and structure
Arrange voxels into a 2d matrix on the xz plane. The x-axis represents column of the matrix and the z-axis represents row of the matrix. The size of matrix is as formula(2).

$$round(range\_Z * k/min + 2) * round(range\_X * k/min + 2) \tag{1}$$

The reason for adding 2 is that points are only distributed around the boundary of the entity and there are no points inside the entity. Therefore,the wrong hollows inside the model will also be determined as hollow bodies. So this paper establishes a layer of voxels outside the minimum cube bounding box and uses the number of voxels in non-model area adjacent with each hollow body to filter out wrong hollow body inside the model.

This paper defines a structure grid for each region of the matrix.Each structure contains an array containing round(range_Y/dt) voxels.The coordinates of the array represent the height information.The schematic diagram is as shown in Figure 3.

Step 3:Calculate position of points
After defining the voxels and the matrix structure, we will calculate the location of each point.Assuming that the coordinates of point i is $(x_i,y_i,z_i)$ and the minimum values in x_axis,y_axis,z_axis are x_min,y_min,z_min. So the row,column,height of voxel which point i is located in are as formula(3),(4),(5).

$$round((z_i - z\_min)/(range\_Z/round(range\_Z/d_t)) + 1 \tag{2}$$

$$round((x_i - x\_min)/(range\_X/round(range\_X/d_t)) + 1 \tag{3}$$

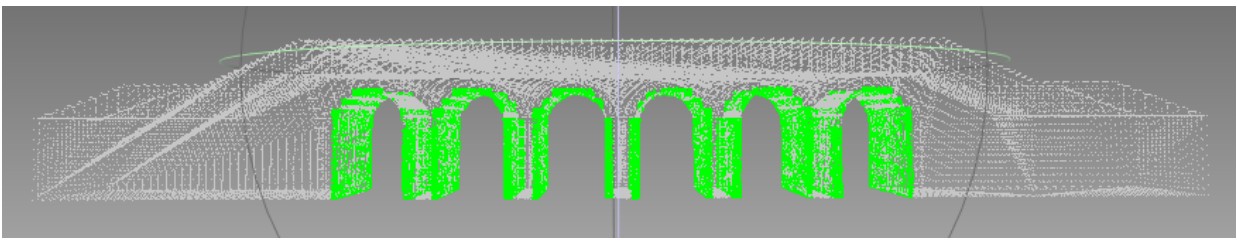

Figure 1: Bridge point cloud data with six hollow bodies

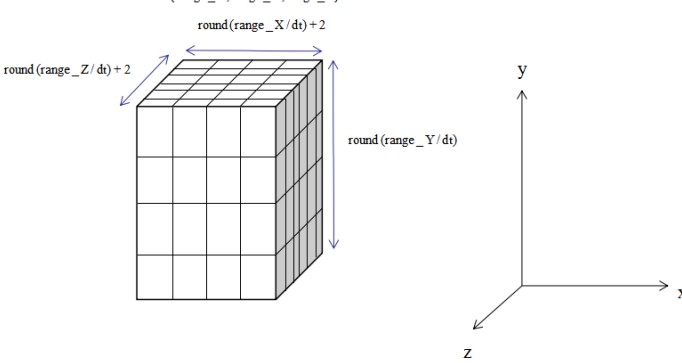

Figure 3: Voxel segmentation of the minimum cube bounding box

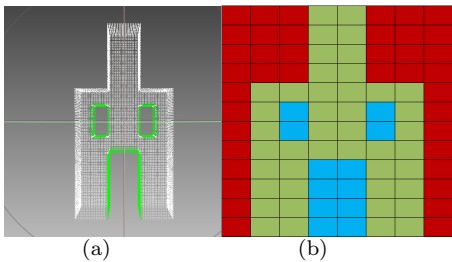

Figure 4: An example of classification of voxels

$$round((y_i - y\_min)/(range\_Y/round(range\_Y/d_t))) \quad (4)$$

### 3.1.3 Determination of the group of hollow body voxels

The rules for judging whether a voxel may belong to a hollow body are as follows:

- The number of points it contains is less than the threshold.

- There exists a voxel that doesn't belong to any hollow body above it.

So this paper first calculates the maximum height of each grid of xz plane matrix. The voxels whose height is lower than the maximum of its region belong to model area and others belong to area outside the model. If a voxel in model area contains less than threshold of points,we will put them in a set Q for further process because they may belong to hollow bodies. Figure 4-b is a schematic diagram of xy plane. The red voxels belong to area outside the model. The green and blue voxels belong to model area and blue voxels will be put in set for segmentation.

### 3.1.4 Segmentation of the group of hollow body voxels

This paper separates the group of voxels based on connectivity. Define the initial group number s as 1 and define a group LK to store the voxels of each group. Then initialize the label of voxels in group as 0.

Step 1:If the label of a voxel in group Q is 0, change its label to s and put it into LK.

Step 2:We define the row,col,height of the voxel is r,c,h. If the voxels whose position are (r+1,c,h),(r-1,c,h), (r,c+1,h),(r,c-1,h),(r,c,h+1),(r,c,h-1) are in Q,change their labels to s and put them into LK.

Step3:Repeat step 2 when new voxels are put into LK until we can't find adjacent voxels.Add s to 1 and clear Lk to store next group of voxels and repeat step1 until all the voxels in Q have been processed.

### 3.1.5 Determination of correctness of hollow bodies

For all groups separated,we use following rules to determine whether these groups are correct.

- Size of hollow body

  Some groups contain a small number of voxels indicating that the proportion of the area is small. So this paper filters out these groups by defining volume ratio between hollow bodies and the model as a threshold.

- Connectivity of hollow body

  The results contains wrong hollow bodies inside the model which is caused by the feature of point cloud. This paper uses the number of non-model voxels adjacent with each hollow body as the other threshold.

The process of step 2 and step 3 is as figure 5.

At last,this paper traverses all the points in the point cloud data to determine the boundaries of each hollow body by the space position.

## 3.2 Hollow body parameter

This paper defines a set of parameters containing volume ratio,normal equation,depth ratio of hollow bodies in order to accurately describe their characteristics for further research. The first phase of the algorithm just need to calculate the volume ratio while the second phase of algorithm calculate all the three parameters.

### 3.2.1 Volume ratio

This paper defines the result that volume of hollow body divides by volume of model as the volume radio in order to describe the size of each hollow body.

We assume the volume of the model is V and the volume of hollow body i is $S_i$,so the volume ratio of hollow body i is $\frac{S_i}{V}$. The parameter ranges from 0 to 1.However we can't calculate the accurate volumes of model and hollow bodies in

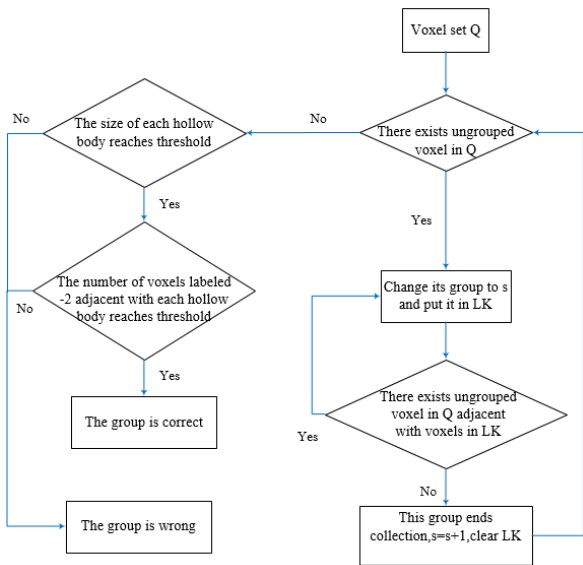

Figure 5: Process of hollow body determination

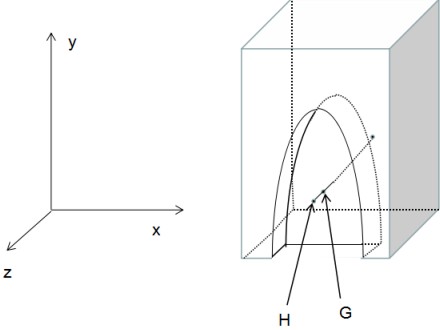

Figure 6: An example of normal line of hollow body

point cloud because of the variability and irregularity of the point cloud data. Therefore,we obtain the approximation of each volume ratio by calculating the result that the number of voxels in each hollow body divide by the number of voxels in model area.

### 3.2.2 Normal equation

This paper defines the normal equation of the hollow body in order to describe the position of each hollow body.

We calculate the normal equation of the hollow body by slice method and it represents the direction of the hollow body extension. For the hollow body i,we first calculate the mean coordinates of the voxels that make up the hollow body and name it as the gravity point $G_i$. We assume the voxel set belonging to hollow body i is $T_i=\{a_1,a_2\ldots a_{n-1},a_n\}$.

In the voxel structure,i represents the row of matrix,j represents the column,h represents the height,$d_i$ represents the spacing of row,$d_j$ represents the spacing of column,$d_h$ represents the spacing of height.The coordinate of $G_i$ is formula(6).

$$( \quad X\_min + (\frac{\sum_{t=1}^{n} a_t.j}{n} - 0.5) * d_j, Y\_min +$$

$$( \quad \frac{\sum_{t=1}^{n} a_t.h}{n} + 0.5) * d_h, Z\_min + (\frac{\sum_{t=1}^{n} a_t.i}{n} - 0.5) * d_i)(5)$$

Then,we use the slice method to slice the yz plane and the xz plane and calculate the row and column of voxels belonging to the hollow body i that appear at the first time or the last time. We assume the voxel set whose row equals the first row of the model area in hollow body i is $R_i=\{r_1,r_2\ldots r_{m-1},r_m\}$ and the coordinate of $H_i$ is as formula(7).

$$( \quad X\_min + (\frac{\sum_{q=1}^{m} r_q.j}{m} - 0.5) * d_j, Y\_min +$$

$$( \quad \frac{\sum_{q=1}^{m} r_q.h}{m} + 0.5) * d_h, Z\_min + 0.5 * d_i) \qquad (6)$$

Otherwise,We assume the voxel set whose row equals the last row of model area in hollow body i is $N_i=\{b_1,b_2\ldots b_{f-1},b_f\}$ and the coordinate of $H_i$ is as formula(8).

$$( \quad X\_min + (\frac{\sum_{q=1}^{f} b_q.j}{f} - 0.5) * d_j, Y\_min +$$

$$( \quad \frac{\sum_{q=1}^{f} b_q.h}{f} + 0.5) * d_h, Z\_min + (w - 0.5) * d_i) \ (7)$$

Similarly,we can also calculate the midpoint of the first column and the last column of the hollow body i. If the hollow body has midpoints in both boundary surfaces,we use the gravity point and the midpoint in the first row or first column of hollow body to calculate the normal equation.

For hollow body i,we use the gravity point $G_i$ and the surface midpoint $H_i$ to calculate the normal line. The equation of the normal line is as formula(9) and the direction vector is as formula(10).

$$\frac{x - H_i.x}{G_i.x - H_i.x} = \frac{y - H_i.y}{G_i.y - H_i.y} = \frac{z - H_i.z}{G_i.z - H_i.z} \qquad (8)$$

$$(G_i.x - H_i.x, G_i.y - H_i.y, G_i.z - H_i.z) \qquad (9)$$

If the hollow body connects with the space outside the model in both directions, we will define the direction that has a bigger depth ratio which is the next parameter as the main direction of the hollow body. The hollow body normal equation can represent the space position of the hollow body. An example is shown in Figure 6. The line through HG is the normal line and the the direction of the normal line is from H to G.

### 3.2.3 Depth ratio

This paper defines depth ratio of each hollow body in order to describe the shape of the hollow body. The parameter ranges from 0 to 1. When the parameter approaches 0,it means the hollow body is very shallow. When the parameter equals 1,it means the hollow body is full penetrative. The calculation of the depth ratio depends on the normal of the hollow body. We assume the maximum of the point cloud in x-axis is x_max,the minimum in x-axis is x_min,the maximum in z-axis is z_max and the minimum in z-axis is z_min.For each hollow normal equation,if it is extended along the z-axis,we set z=z_min and z=z_max and calculate the distance $S_{maxi}$ between the two points. And then we

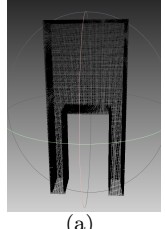
(a)
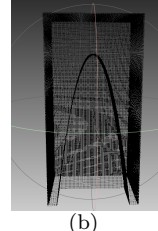
(b)
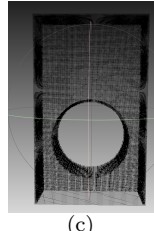
(c)

Figure 7: Test data

set formula(11) and calculate the distance $S_{row}$ between the two points.

$$
\begin{aligned}
z &= z\_min + (first\_row - 1) * d_i \\
z &= z\_min + last\_row * d_i \qquad (10)
\end{aligned}
$$

If the normal is extended along the x-axis,we set x=x_min and x=x_max and calculate the distance between the two points $S_{maxj}$. And then we set formula(12) and calculate the distance between the two points $S_{col}$.

$$
\begin{aligned}
x &= x\_min + (first\_col - 1) * d_j \\
x &= x\_min + last\_row * d_j \qquad (11)
\end{aligned}
$$

If both directions have a normal line,we will take max $\{\frac{S_{row}}{S_{maxi}}, \frac{S_{col}}{S_{maxj}}\}$ as depth ratio of the hollow body and the normal line in this direction will be the normal of the hollow body. If there only exists a normal line in one direction,then the corresponding distance ratio $\frac{S_{row}}{S_{maxi}}$ or $\frac{S_{col}}{S_{maxj}}$ will be the depth ratio of the hollow body.

## 4    Experiment

We can't calculate the accurate volumes of model and hollow bodies in point cloud because of the variability and irregularity of the point cloud data. Therefore,this paper chooses to establish several sets of point cloud data with different sizes and shapes of regular hollow bodies. We calculate their accurate volumes to test the optimal number of voxels to process the point cloud data with different sizes of hollow bodies.

We divide the test data into three groups according to the shape of the hollow body.The schematic diagram of the test data is shown in Figure 7. After the calculation of different values of k,the results are counted. The experiment finds that when the number of voxels that divide the model is close to 24000, 60000, 100000, 150,000,and 200000, the results are more representative. The experimental results are shown in Figure 8.

After analyzing the experimental results,we find that the relationship between the number of voxels and the volume ratio V can be defined as a discrete function when we want to achieve 95% correct rate of the volume ratio as formula(15).

$$
S = \begin{cases}
24000 & V \geq 0.5 \\
60000 & 0.35 \leq V < 0.5 \\
100000 & 0.25 \leq V < 0.35 \\
150000 & 0.1 \leq V < 0.25 \\
200000 & V < 0.1
\end{cases} \qquad (12)
$$

The different choices for each interval can make the algorithm achieve the correct rate of more than 95%. If we add to the number of voxels,the correct rate will not have a significant improvement but the processing time will be greatly

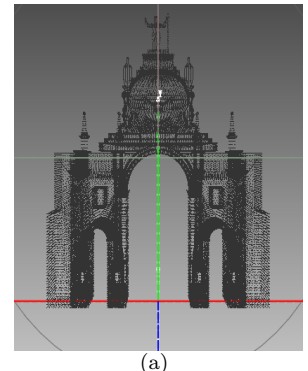
(a)
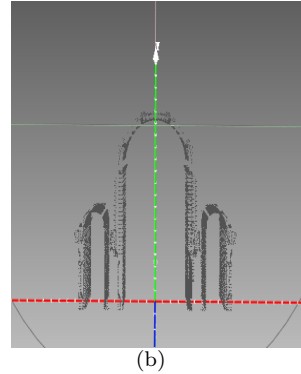
(b)

**Results of first phase: 24000 voxels**

row:15          column:52          H:56
Number of voxels in model area: 22416
Number of voxels in non-model area: 21264

Hollow body 1
Number of voxels in hollow body 1: 575
Volume ratio: 0.0256513

Hollow body 2
Number of voxels in hollow body 2: 4970
Volume ratio: 0.221717

Hollow body 3
Number of voxels in hollow body 3: 575
Volume ratio: 0.0256513

(c)

Figure 9: First phase result of European building point cloud

increased. So we use this discrete function as the basis for the subsequent determination of the hollow bodies.

For the data to be processed,we will first set the initial value of the number of voxels to 24000 and calculate the approximate value of the volume ratio of each hollow body. Then we use the smallest ratio and point cloud density which must ensure the distance between adjacent points is bigger than the length of side of voxels to calculate the optimal number of voxels that divide the point cloud. Figure 9 is the result of first phase of European building data and figure 10 is the second phase result.

Considering the concept of hollow body is proposed in this paper for the first time, we can't make comparision with previous research. Therefore,this paper establishes a data set of 287 point cloud data from various 3D model websites and 3d reconstruction to verify the effect of the algorithm. The data set can be divided into seven types containing natural landscape,metal component,indoor furniture,jewelry sculpture,bridge,building and test data. Because there are usually only a few key points in a surface in 3d models,this paper uses the Loop subdivision surface [19] and the Catmull-Clark subdivision surface [20] through meshlab to achieve the upsampling of the 3D model to form a point cloud file. We will find more point cloud data with hollow bodies in further work and the website is under construction.

After experimenting with the 287 experimental data,this paper compares the results with the calibrated hollow bodies. The results show that 236 data are successfully judged and 51 data fail.The correct rate is 82.2%.Some results are

| shape of surface of hollow body | acurate volume ratio | correct rate of 24000 voxels | correct rate of 60000 voxels | correct rate of 100000 voxels | correct rate of 150000 voxels | correct rate of 200000 voxels |
|---|---|---|---|---|---|---|
| rectangle | 0.72 | 95.00% | 96.88% | 97.26% | 98.01% | 98.35% |
| rectangle | 0.48 | 94.44% | 95.83% | 97.14% | 96.06% | 96.77% |
| rectangle | 0.3 | 93.47% | 95.13% | 96.67% | 95.88% | 96.00% |
| rectangle | 0.2 | 91.67% | 93.75% | 95.31% | 97.33% | 96.87% |
| rectangle | 0.12 | 91.16% | 93.47% | 95.17% | 97.02% | 96.77% |
| rectangle | 0.05 | 90.34% | 91.93% | 95.02% | 95.12% | 95.04% |
| circular | 0.53305 | 96.59% | 97.56% | 98.64% | 99.23% | 99.12% |
| circular | 0.32725 | 92.24% | 93.90% | 96.18% | 96.69% | 97.40% |
| circular | 0.20944 | 91.07% | 93.25% | 94.61% | 95.35% | 96.46% |
| circular | 0.11781 | 88.03% | 92.40% | 93.79% | 95.53% | 95.89% |
| circular | 0.05236 | 81.17% | 90.02% | 91.13% | 93.09% | 95.11% |
| curve surface | 0.5333 | 96.51% | 97.11% | 97.48% | 98.02% | 98.53% |
| curve surface | 0.36 | 92.59% | 95.68% | 96.22% | 96.28% | 97.64% |
| curve surface | 0.25 | 91.74% | 93.77% | 95.08% | 95.64% | 96.26% |
| curve surface | 0.19753 | 91.18% | 93.15% | 94.11% | 95.21% | 95.43% |
| curve surface | 0.08 | 82.43% | 87.16% | 90.73% | 92.86% | 95.01% |

Figure 8: Experimental results of different number of voxels

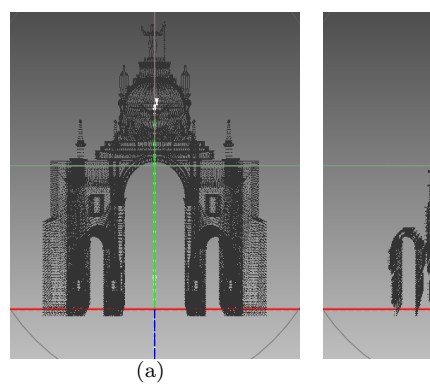 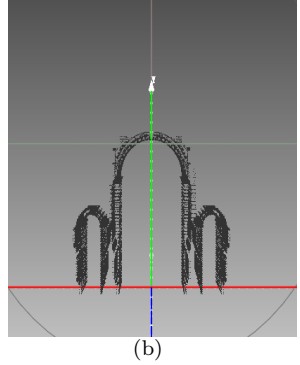

(a)     (b)

Results of second phase: 200000 voxels

row:31  column:111  H:121
Number of voxels in model area: 216354
Number of voxels in non-model area: 200007

Hollow body 1

Normal equation: $\dfrac{x+144.628}{0.144135}=\dfrac{y-75.4716}{-3.06932}=\dfrac{z-70.5837}{63.6174}$

Number of voxels in hollow body 1: 8400
Volume ratio: 0.0388253
Depth ratio in z direction: 1

Hollow body 2

Normal equation: $\dfrac{x-0.0778378}{0.0936333}=\dfrac{y-146.945}{-3.03587}=\dfrac{z-70.5837}{63.6095}$

Number of voxels in hollow body 2: 53640
Volume ratio: 0.247927
Depth ratio in z direction: 1

Hollow body 3

Normal equation: $\dfrac{x-144.628}{-0.016098}=\dfrac{y-75.4716}{-3.00755}=\dfrac{z-70.5837}{63.6005}$

Number of voxels in hollow body 3: 8392
Volume ratio: 0.0387883
Depth ratio in z direction: 1

(c)

Figure 10: Second phase result of European building point cloud

as figure 11 12.

There are several reasons why the algorithm fails to process:

- For some data with complex components such as patterns,the density of points in the pattern part is high and this will increase the density of the whole point cloud.So it may cause the algorithm to cause errors when estimating the optimal number of voxels that divide the minimum cube bounding box.

- For the point cloud data obtained through 3D reconstruction,if the effect reconstruction is bad causing large holes appeared in the point cloud,the algorithm will fail to determine the correct hollow body.

## 5 Conclusion

Considering the display limitation of volumetric 3d display,this paper proposes the concept of hollow body. This paper gives a quantitative description of hollow body and defines a set of parameters containing volume ratio,normal line,depth ratio to describe their size,position and shape. Then this paper proposes a novelty VCRHD algorithm to accurately determine the hollow bodies in point cloud. Because there aren't related work to deal with the hollow body before, this paper establishes a data set containing 287 different point cloud files to test algorithm and to be used as a reference for follow-up studies. The algorithm may be affected by the noisy points and holes in the point cloud which need to be improved in further research.

Accurately defining the hollow body can help subsequent work. Further reserach can realize a more realistic three-dimensional display by simulating the visual experience observed by the human eye through the hollow body and can help in CAD or surface construction. We hope researchers in these areas can benefit from the application of hollow body.

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

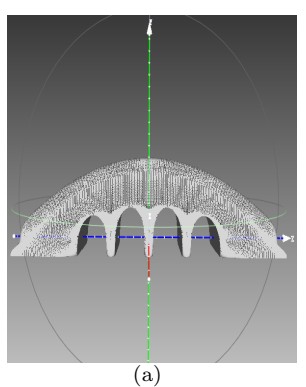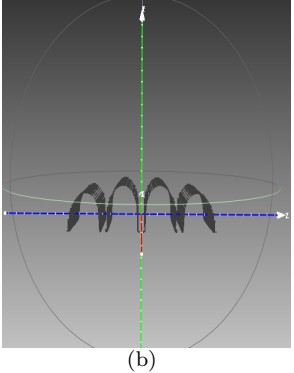

(a)     (b)

Hollow body1

Normal equation: $\dfrac{x+79.4118}{-79.4084}=\dfrac{y-7.16}{-0.0750799}=\dfrac{z+134.468}{-0.0985718}$

Depth ratio in x direction: 1
Volume ratio: 0.0577731

Hollow body2

Normal equation: $\dfrac{x+79.4118}{-79.4047}=\dfrac{y-13.4512}{-0.13721}=\dfrac{z+44.4676}{0.01091}$

Depth ratio in x direction: 1
Volume ratio: 0.0705077

Hollow body3

Normal equation: $\dfrac{x+79.4118}{-79.4047}=\dfrac{y-13.4512}{-0.13721}=\dfrac{z-44.4677}{0.0149612}$

Depth ratio in x direction: 1
Volume ratio: 0.0705077

Hollow body4

Normal equation: $\dfrac{x+79.4118}{-79.4084}=\dfrac{y-7.16}{-0.0750799}=\dfrac{z-134.468}{0.130142}$

Depth ratio in x direction: 1
Volume ratio: 0.0577731

(c)

Figure 11: Bridge data

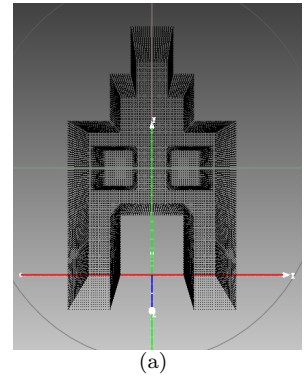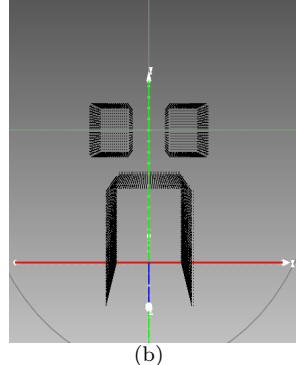

(a)     (b)

Hollow body1

Normal equation: $\dfrac{x+2}{0.000385523}=\dfrac{y-6}{0.00154257}=\dfrac{z-0.090909}{-2.90903}$

Depth ratio in z direction: 0.75
Volume ratio: 0.0321612

Hollow body2

Normal equation: $\dfrac{x-0.0909092}{0.000116177}=\dfrac{y-2}{0}=\dfrac{z-0.090909}{-3.90908}$

Depth ratio in z direction: 1
Volume ratio: 0.198113

Hollow body3

Normal equation: $\dfrac{x-2}{0.00159812}=\dfrac{y-6}{0.00154257}=\dfrac{z-0.090909}{-2.90903}$

Depth ratio in z direction: 0.75
Volume ratio: 0.0321612

(c)

Figure 12: Building data

[4] Yuan Liang:Construction of 3D Entity Display System for data Processing And Virtual Display.In:Tianjin University,2017.

[5] Lee JS,Lee MY,Kim JO,Kim CJ,Won YH:Novel volumetric 3D display based on point light source optical reconstruction using multi focal lens array.In:Conference on Advances in Display Technologies V,FEB,2015.doi: 10.1117/12.2078101

[6] Jun Sun:Voxel Analysis and Software Design Based on Array LED 3D Display System.In:Huazhong University of Science and Technology,2017.

[7] Orriols X,Binefa X:Finding breaking curves in 3D surfaces.LECTURE NOTES IN COMPUTER SCIENCE 2652(2003),681-688.

[8] Yongtae Jun:A piecewise hole filling algorithm in reverse engineering.Computer-Aided Design,37(2005),263C270.doi: 10.1016/j.cad.2004.06.012

[9] Jia Li,Huan Lin,Duo Qiang Zhang,Xiao Lu Xue.Extracting geometric edges from 3D point clouds based on normal vector change.Applied Mechanics and Materials.2014(571):729-734.doi: 10.4028/www.scientific.net/AMM.571-572.729

[10] Xianfeng Huang,Xiaoguang Cheng,Fan Zhang,GONG Jianya.Side ratio constrain based precise boundary tracing algorithm for distance point clouds[C].The International Archives of the Photogrammetry,Remote Sensing and Spatial Information Sciences,Beijing 2008.

[11] Emelyanov Alexander,Skala Vclav.Surface reconstruction from problem point clouds.Virtual Environment on PC Cluster 2002, workshop proceedings,Russia,p:68-75.doi: 11025/11694

[12] Soo-Kyun Kim:Extraction of ridge and valley lines from unorganized points.Multimedia Tools and Applications,2013,63(1),265-279.doi: 10.1007/s11042-012-0999-y

[13] Kurlin Vitaliy.A fast and robust algorithm to count topologically persisitent holes in noisy clouds.27th IEEE Conference on Computer Vision and Pattern Recognition(CVPR),JUN 23-28,2014.doi: 10.1109/CVPR.2014.189

[14] Van Sinh Nguyen,Trong Hai Trinh,Manh Ha Tran.Hole Boundary Detection of a Surface of 3D point clouds.2015 International Conference on Advanced Computing and Applications.doi: 10.1109/ACOMP.2015.12

[15] Tengfei Bao,Jinlei Zhao,Miao Xu.step edge detection method for 3D point clouds based on 2D range images.Optik 126 (2015) 2706C2710.

[16] Zhenqing Yang,Yonglei Yong.Boundary extraction based on point cloud slices[J].Computer Applications and Software,2014,**31**(1):222-224+245.

[17] Shaoyan Gai,Feipeng Da,Lulu Zeng,Yuan Huang.Research on a hole filling algorithm of a point cloud based on structure from motion.Journal of the Optical Society of America A **36**(2),Feb,2019.doi: 10.1364/JOSAA.36.000A39

[18] Fan LU,Song Li,Jingjing Cao,Xiaozheng E,Yong Zhou:Algorithm for extraction of point cloud boundary point based on inverse distance weight and density.COMPUTER ENGINEERING AND DESIGN **40**(2),364-369(2019).doi: 10.16208/j.issn1000-7024.2019.02.012

[19] Li Zhang,Xiangrong She,Xianyu Ge,Jieqing Tan:Adaptive fitting algorithm of progressive interpolation for Loop subdivision surface.International Journal of distributed sensor networks **14**(11),Nov.2018.doi: 10.1177/1550147718812355

[20] Shuqun Liu,Bei Zhang:Mathematical Model of Catmull-Clark Subdivision Scheme on Regular Mesh.In:Proceedings of 2017 2nd International Conference on Automation,Mechanical Control and Computational Engineering,March,2017.doi: 10.2991/amcce-17.2017.31
