# OpenReview forum: "Determination and quantitative description of hollow body in point cloud"
_graphicsinterface.org/Graphics_Interface/2020/Conference — Submitted to GI 2020_

### Official Review · AnonReviewer2 · 2020-04-13
**Unclear motivation and contribution**

**Rating:** 2
**Confidence:** 5

**Review:**

The paper presents a method to find "hollow bodies" in point clouds by voxelizing the space of point clouds and the adjacency information of the voxels. For the following reasons, I suggest to reject the paper:

1) The motivation of the paper is questionable. The main motivation discussed in the introduction is to use the proposed method to improve the quality of 3D displays. However, no visual result of such a system is presented in the paper to back up this claim.

2) The voxelization technique in the paper, while it seems like a reasonable method, is straightforward. There is no clear contribution in the proposed method for the area of Computer Graphics.

3) There is no clear mathematical definition for hollow bodies. "Above, upper, and outer" terminologies in the definition of the hollow body make this definition unclear and coordinate dependent.  I was looking for a better mathematical definition which could be considered as a contribution.

4) The results' rendering and presentation are poor.

5) The paper is full of grammar mistakes and typos. I suggest that the paper gets thoroughly edited by an English native speaker. These are *some* of the *many* mistakes:

*) space is needed after comma, periods, etc.
*) Page 1: "-This paper proposes a novelty voxel connectivity" novelty---> novel
*) Page 1: "Yongtae Jun [ 8 ] ﬁrstly determine" determine---> determines. It is a single author paper.
*) Page 1: "The human brain need to" need----> needs
*) Page 2: "depth ration to describe their" ration---> ratio
*) Figure 2: All the sentences need to be re-written.
*) Page 3: "model as the volume radio in order" radio---> ratio
.....

---

### Official Review · AnonReviewer1 · 2020-04-13
**Problem statement vague; Motivation unclear; Algorithm simplistic**

**Rating:** 3
**Confidence:** 4

**Review:**

The problem statement is unclear and informal throughout most of the paper. Towards the end the reader gets an idea in retrospect, but the involved concepts remain vague. The definition of "hollow body" is essentially via an algorithm (that involves quite a number of rather arbitrary engineering choices) rather than via a mathematical concept (that would then be realized or approximated by an algorithm / implementation).

The motivation remains unclear. No plausible use case for the orientation dependent definition of "hollow body" is given. Then paper mentions (quite extensively!) 3D display technology as well as CAD systems, but ultimately no connection to these is made.

There are numerous language and grammar errors. At times this makes it hard to even make sense of the text, or to be certain things are understood as intended.

The paper spells out algorithmic parts in detail that are standard knowledge (like determination of connected components in a labeled voxel grid) and therefore could be reduced to a short sentence (and perhaps a reference).

Robustness (with respect to non-uniforn sampling, noise, etc.) is not addressed or discussed. Presented results apparently use synthetic point clouds rather than real-world data.

The algorithm makes use of multiple thresholds. Their choice is not discussed. This, together with the various formal clarity issues, hinders reproducibility of the method and the results. In the result section successes and fails are reported; it remained unclear what even constitutes a "fail".

Overall, the paper's presentation is of a quality and clarity that is not acceptable for publication. The paper's contribution is quite limited and its potential significance remained open.

---

### Official Review · AnonReviewer3 · 2020-04-20
**The submission is not clear enough to be evaluated.**

**Rating:** 2
**Confidence:** 3

**Review:**

This submission addresses (I believe...) the problem of finding empty regions in pointsets in a specific setup.

The presented technique relies on a voxelization of the input pointset, and defines a hollow voxel as "a voxel that is closed to the upper area in y-axis (has voxels above itself) and connects with the outer area in x-axis or z-axis". The set of hollow regions is depicted in Fig 2b in blue. The hollow regions are then partitioned into connected components, that finally define the hollow bodies. Each hollow body is then characterized by volume, "normal line" (I do not understand the definition of the normal line as described in the manuscript but I assume, based on Fig6, that it could be the direction of smallest variance given by the PCA passing through the center of mass?) and "depth ratio" (I did not understand this definition either...). The characteristics of the hollow bodies are not really used in the present work, but "could be used for further research".

I am actually not entirely sure of what is described in the manuscript, because I have extreme difficulties in making sense of the phrasing... However, if the authors address the consistent volumetrization of pointsets (or consistent in/out segmentation), there are a number of existing techniques that the authors could compare to. For example:
A Global Parity Measure for Incomplete Point Cloud Data, Seversky and Yin, Pacific Graphics 2012 (does not assume that the input pointset contains normals).

Maybe there are extremely specific constraints that guide the proposed approach, which make the technique not possible to compare to existing work, but then I did not understand them... (and it may be partly my fault).

I believe that given the current state of the manuscript in terms of clarity of exposition, this submission should be rejected. I encourage the authors to proof-read the manuscript before submitting it again.

---

### Meta-Review · Area_Chair1 · 2020-04-22

**Recommendation:** Reject
**Confidence:** 5

**Metareview:**

All reviewers agree that there are major clarity issues in this paper. The concept of "hollow body", which is the central point of this paper, is defined only informally. The motivation for this definition remained unclear. The overall motivation of the paper is partly unclear, partly questionable. The technical and algorithmic aspects are mostly straightforward, no significant novel insights are provided; it remained unclear what could be considered the actual contribution. Reproducibilty is limited, because parameter choices are not discussed.

---

### Decision · Program_Chairs · 2020-04-25

Reject